# Subsidiary Prototype Alignment
# for Universal Domain Adaptation

**Jogendra Nath Kundu**[1*]    **Suvaansh Bhambri**[1*]    **Akshay Kulkarni**[1*]    **Hiran Sarkar**[1]
**Varun Jampani**[2]    **R. Venkatesh Babu**[1]

[1]Indian Institute of Science    [2]Google Research

## Abstract

Universal Domain Adaptation (UniDA) deals with the problem of knowledge transfer between two datasets with domain-shift as well as category-shift. The goal is to categorize unlabeled target samples, either into one of the "known" categories or into a single "unknown" category. A major problem in UniDA is negative transfer, *i.e.* misalignment of "known" and "unknown" classes. To this end, we first uncover an intriguing tradeoff between negative-transfer-risk and domain-invariance exhibited at different layers of a deep network. It turns out we can strike a balance between these two metrics at a mid-level layer. Towards designing an effective framework based on this insight, we draw motivation from Bag-of-visual-Words (BoW). Word-prototypes in a BoW-like representation of a mid-level layer would represent lower-level visual primitives that are likely to be unaffected by the category-shift in the high-level features. We develop modifications that encourage learning of word-prototypes followed by word-histogram based classification. Following this, subsidiary prototype-space alignment (SPA) can be seen as a closed-set alignment problem, thereby avoiding negative transfer. We realize this with a novel word-histogram-related pretext task to enable closed-set SPA, operating in conjunction with goal task UniDA. We demonstrate the efficacy of our approach on top of existing UniDA techniques[1], yielding state-of-the-art performance across three standard UniDA and Open-Set DA object recognition benchmarks.

## 1 Introduction

Despite the success of deep networks trained on large-scale datasets, they are found to be brittle under input distribution shift *i.e.* domain-shift [6]. Thus, adapting a trained model for a new target environment becomes challenging as data annotation is too expensive or time-consuming [7] for every new target dataset. Unsupervised Domain Adaptation (DA) [11] is one of the solutions to this problem where knowledge is transferred from a labeled source domain to an unlabeled target domain.

While most works [12, 25, 19] focused on Closed-Set DA, where source and target label sets are shared ($\mathcal{C}_s = \mathcal{C}_t$), recent works introduced disjoint label set scenarios like Partial DA [51, 4] ($\mathcal{C}_t \subset \mathcal{C}_s$) and Open-Set DA [37, 17] ($\mathcal{C}_s \subset \mathcal{C}_t$). However, the most practical setting is Universal DA (UniDA) [48] where the relation between the source and target label sets is unknown *i.e.* with any number of shared, source-private and target-private classes. In UniDA, a model is trained to categorize unlabeled target samples into one of the shared classes ("known" classes) or into a single "unknown" class.

The major problem in UniDA is negative-transfer [38] where misalignment between the shared and private classes degrades the adaptation performance. On the other hand, we have the domain-shift problem, which is usually handled by learning domain-invariant features [48, 18, 21, 34]. So, we first

---

[*]equal contribution
[1]Project Page: https://sites.google.com/view/spa-unida

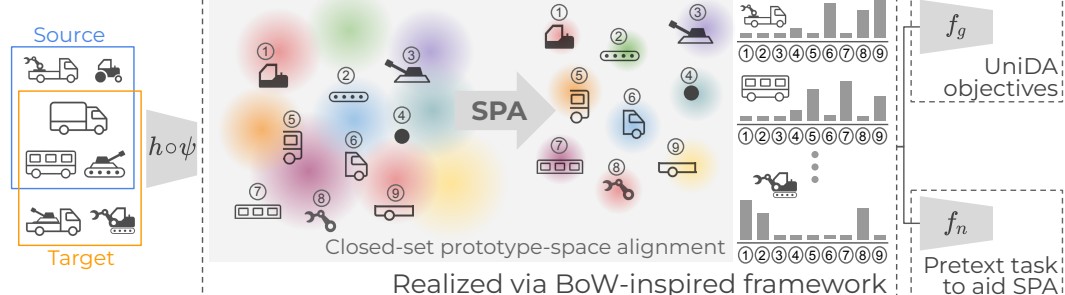

Figure 1: UniDA involves source and target data with shared and private classes. The colored blobs represent target features w.r.t. the word-prototypes (numbered) that represent lower-level visual primitives. With a word-related pretext task guiding the Subsidiary Prototype-space Alignment (SPA) to avoid word-level misalignment, the word-histogram output-space can better represent the intrinsic class-structure (including unknown classes), leading to better UniDA performance.

perform a control experiment to analyze the negative-transfer-risk (NTR) and domain-invariance-score (DIS) at different layers of the deep network (Fig. 2). NTR is measured through the class-specificity of the features via an entropy-based shared-vs-unknown binary classifier while DIS is measured as the inverse of the standard $\mathcal{A}$-distance [1] that quantifies domain discrepancy. We observe that NTR and DIS are at odds with each other *i.e.* NTR increases while DIS decreases as we move from lower to higher-level (deeper) layers. Thus, we arrive at contradicting solutions where avoiding negative-transfer requires adaptation at a lower-level layer while effective domain-invariance requires adaptation at a deeper layer. While a balance can be naively struck at a mid-level layer, we ask, can we further develop and constrain the mid-level representation space to avoid negative transfer?

Motivated by Bag-of-visual-Words (BoW) [49], we hypothesize that a BoW-like representation at a mid-level layer would represent lower-level visual primitives that are unaffected by the category-shift in the higher-level features. We illustrate this idea in Fig. 1 with a UniDA scenario where source and target have private and shared classes. Note that, in most practical scenarios, the private categories are usually related to the shared categories (*e.g.* a tractor and an excavator may be private classes in rural and urban scenes respectively, but they share some common visual attributes like their chassis). Thus, in a word-prototype-space, different visual primitives can be shared across domains and classes (including unknown classes) and a closed-set alignment of target features with the primitives can be performed. Next, we explain how to realize this subsidiary prototype-space alignment (SPA).

First, we propose architecture modifications to introduce explicit word-prototypes and extract a word-histogram output. We analyze the alignment of feature vectors with different word-prototypes. Here, better prototype-space alignment would imply sparser word-histograms as a specific word-prototype would have a significant contribution to the word-histogram w.r.t. other prototypes. Intuitively, with higher sparsity, the word-histogram space can better represent the intrinsic class structure (including unknown classes). To enforce this sparsity *i.e.* the subsidiary prototype-space alignment (SPA), we minimize a self-entropy loss at the word-histogram level. However, this remains susceptible to word-level misalignment due to a lack of word-related priors.

Thus, we seek a word-related prior that can be cast into a self-supervised pretext task so that both labeled source and unlabeled target can be used. Given that each instance can be represented as a word-histogram, we find a simple property based on grid-shuffling of image crops (Fig. 4). Here, the word-histogram entropy of a grid-shuffled image increases with the number of distinct instances that contribute crops. Based on this, we create a novel pretext task to classify the number of instances used in an input grid-shuffled image. Intuitively, it encourages prototype alignment (SPA) as distinguishing different classes in this pretext task becomes easier with higher sparsity of word-histograms.

To summarize, our contributions are as follows,

- We are the first to uncover and analyze the tradeoff between negative-transfer and domain-invariance in UniDA. While a naive balance can be struck, we introduce BoW-inspired word-prototypes and a subsidiary prototype-space alignment (SPA) objective to further alleviate negative-transfer.
- We devise a word-related prior and cast it as a self-supervised pretext task to further improve SPA by avoiding word-level misalignment.
- Our approach, coupled with existing UniDA approaches, yields state-of-the-art performance across three standard Open-Set DA and UniDA benchmarks for object recognition.

## 2 Related work

**Open-Set DA (OSDA)** has been studied in several scenarios [37, 29, 3, 8]. We focus on the case given by [37], where target domain contains private classes, unknown to source. [37] presented an adversarial learning method where feature generator learns known-unknown separation. Other works focus on anomaly measurement [30] or learn to discriminate known and unknown samples [47, 33, 28]. ROS [3] uses rotation prediction to separate known and unknown samples whereas our word-based pretext task regularizes UniDA and implicitly improves known-unknown separation.

**Universal DA (UniDA)** [48] is a complex DA scenario that assumes no prior knowledge of the relationship between the source and target label spaces. Similar to Open-Set DA, UniDA also requires identification of target-private classes. Prior works [10, 48, 38, 3] computed a confidence score for known classes, and data with a score below a threshold were considered unknown. [38] proposed neighbourhood clustering to understand the target domain structure and an entropy separation loss for feature alignment. OVANet [36] used binary classifiers in a one-vs-all manner to identify unknown samples, and DCC [23] used domain consensus knowledge to find discriminative clusters in both shared and private data. In contrast to prior arts, we draw motivation from Bag-of-visual-Words and construct a pretext task to complement these works by enhancing their intrinsic domain structure.

**BoW-related works.** Early works utilized Bag-of-visual-Words (BoW) representations for downstream applications like object recognition [44], object detection [2], image retrieval [39], *etc*. More recent work [13, 14] leveraged BoW prediction for self-supervised learning of representations for downstream tasks. To the best of our knowledge, we are the first to utilize BoW concepts in UniDA.

## 3 Approach

### 3.1 Preliminaries

Consider a labeled source dataset $\mathcal{D}_s = \{(x_s, y_s) : x_s \in \mathcal{X}, y_s \in \mathcal{C}_s\}$ where $\mathcal{C}_s$ is the source label set, $\mathcal{X}$ is the input space, and $x_s$ is drawn from the marginal distribution $p_s$. Also consider an unlabeled dataset $\mathcal{D}_t = \{x_t : x_t \in \mathcal{X}\}$ where $x_t$ is drawn from the marginal distribution $p_t$. Let $\mathcal{C}_t$ denote the target label set. In Universal DA [48], the relationship between $\mathcal{C}_s$ and $\mathcal{C}_t$ is unknown. Without loss of generality, the shared label set is $\mathcal{C} = \mathcal{C}_s \cap \mathcal{C}_t$ and the private label sets for source and target are $\overline{\mathcal{C}}_s = \mathcal{C}_s \setminus \mathcal{C}_t$ and $\overline{\mathcal{C}}_t = \mathcal{C}_t \setminus \mathcal{C}_s$ respectively. Next, we define two measures for our insights.

**Negative-Transfer-Risk (NTR).** Negative transfer [43] is a major problem in DA where class-level misalignment occurs (*e.g.* source class "dog" may get wrongly aligned with target class "cat" due to some similarities between the two classes). The problem is aggravated in UniDA as both source and target may have private classes which may be wrongly aligned with the shared classes [48]. Thus, we introduce a *negative-transfer-risk* (NTR) $\gamma_{NTR}(h)$ for a given feature extractor $h : \mathcal{X} \to \mathcal{Z}$, where $\mathcal{Z}$ is an intermediate feature-space. NTR is computed as the target shared-vs-private classification accuracy via a self-entropy threshold on a source-trained linear task-classifier (see Suppl. for complete details),

$$\gamma_{NTR}(h) = \mathbb{E}_{(x_t, y_t^{(sp)}) \sim \mathcal{D}_t} \mathbb{1}(\hat{y}_t^{(sp)}, y_t^{(sp)}) \tag{1}$$

Here, $\hat{y}_t^{(sp)}$ is the prediction (shared or private) using a fixed self-entropy threshold and $y_t^{(sp)}$ represents shared-private label (0 for shared, 1 for private). We access the shared-private labels for a subset of target data only for analysis (not for training). Intuitively, if the features $h(x)$ are highly class-specific, then target-private samples would yield highly uncertain predictions (as they are unseen by the source-trained linear classifier) compared to shared-label samples. Thus, target-private samples would be easily separable with the self-entropy threshold, leading to a high NTR.

**Domain-Invariance-Score (DIS).** The standard $\mathcal{A}$-distance [1] measures the discrepancy between two domains. It is computed using the accuracy of a linear domain classifier (source vs target). Since we aim to measure domain invariance, *i.e.* the inverse of domain discrepancy, we use the inverse of $\mathcal{A}$-distance between the source and target datasets for the given feature extractor $h$,

$$\gamma_{DIS}(\mathcal{D}_s, \mathcal{D}_t) = 1 - \frac{1}{2} d_{\mathcal{A}}(\mathcal{D}_s, \mathcal{D}_t) \tag{2}$$

Here, $d_{\mathcal{A}}(.,.)$ denotes $\mathcal{A}$-distance computed using feature outputs of $h$. Note that $0 \leq d_{\mathcal{A}}(.,.) \leq 2$.

## 3.2 Balancing negative-transfer-risk and domain-invariance

Given that UniDA is highly susceptible to negative-transfer due to the category-shift problem, we analyze the negative-transfer-risk at different layers of the deep network. Here, we consider that shallow layers encode low-level visual features like edges, corners, *etc.* while deeper layers encode more abstract, class-specific features [50]. In the context of UniDA, the feature space of the deeper layers for source and target would be more difficult to align due to the disjointness of the source and target label sets. We empirically observe the same, *i.e.* NTR increases as we go deeper in the model (solid blue curve in Fig. 2). This suggests that adaptation should be performed at a shallower layer.

While we have considered the category-shift problem to understand where to perform the adaptation, we cannot overlook the domain-shift problem. Most DA works [12] perform adaptation at deeper layers, which tend to learn increasingly more domain-specific features [40, 20]. The higher capacity of the deeper layers leads to unregularized domain-specific learning. We also empirically observe that DIS (inverse of domain-specificity) decreases for the deeper layers (solid pink curve in Fig. 2). This suggests that adaptation should be performed at a deeper layer to encourage domain-invariance.

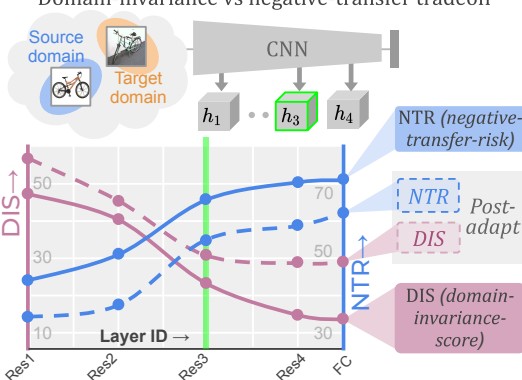

Figure 2: Negative-transfer-risk (NTR) ($\downarrow$) increases in deeper layers while domain-invariance (DIS) ($\uparrow$) decreases. Thus, we adapt at a mid-level layer, where NTR reduces and DIS increases.

Thus, the problems of domain-shift and category-shift are contradictory in suggesting adaptation at a deeper and shallower layer respectively. In other words, there is a tradeoff between DIS and NTR (Fig. 2) where reducing the negative-transfer-risk negatively affects the domain-invariance and vice versa. Thus, we define a criterion for minimal negative-transfer and maximal domain-invariance.

**Definition 1. (Optimal tradeoff between negative-transfer and domain-invariance)** *Consider that adaptation is performed at the $L^{th}$ layer of the backbone $h$ (let $h_L$ denote the backbone upto $L^{th}$ layer). Then adaptation at $h_L$ will encounter minimal negative-transfer and exhibit maximum domain-invariance if with at least $(1-\delta)$ probability, $\gamma_{NTR}(h_L)$ does not exceed $\zeta_n$ by more than $\varepsilon_n$, and $\gamma_{DIS}(h_L)$ exceeds $\zeta_d$ by no less than $\varepsilon_d$, i.e.,*

$$\mathbb{P}[(\gamma_{NTR}(h_L) \leq \zeta_n + \varepsilon_n) \cap (\gamma_{DIS}(h_L) \geq \zeta_d - \varepsilon_d)] \geq 1 - \delta \tag{3}$$

Thus, an optimal tradeoff requires NTR to be less than the threshold $\zeta_n$ and DIS to be greater than the threshold $\zeta_d$ simultaneously. Empirically, we find a good tradeoff at a mid-level layer (green vertical line in Fig. 2, *e.g.* Res3 block in ResNet), *i.e.* a compromise between the contradicting suggestions. Note that we re-use $h$ to represent the backbone upto an optimal layer $L$ instead of $h_L$ for simplicity.

**Why is low NTR desirable?** High NTR implies known and unknown samples are well-separated. However, unsupervised adaptation at a higher NTR layer is more susceptible to misalignment between shared and private (unknown) classes because target-private classes get grouped into a single unknown cluster. In contrast, lower NTR feature space can better represent all the different classes without grouping the target-private classes into a single unknown cluster (*i.e.* better intrinsic structure). Hence, alignment in this space would better respect the separations of private classes than at a higher NTR feature space, which is necessary to avoid misalignment in UniDA. For example, consider "hatchback" (compact car) and "SUV" (large-sized car) as a shared and target-private class, respectively, in an object recognition task. Before adaptation, at a high NTR layer, hatchback and SUV features would be well-separated as SUV is yet unseen to the source-trained model. However, during unsupervised adaptation, the similarities between hatchbacks and SUVs may align the single target-private cluster (containing SUV features) with the hatchback cluster. Due to this, other target-private classes also become closer to this hatchback class which increases the misalignment. In contrast, at a lower NTR layer, the target-private classes (including SUV) would not be grouped together. Hence, misalignment of hatchback and SUV clusters would not disturb other clusters unlike the higher NTR scenario.

### 3.3 Conceptualizing Bag-of-visual-Words (BoW) for UniDA

Now we ask whether a meaningful representation space can be developed at this mid-level layer to effectively mitigate negative-transfer. To this end, we draw motivation from traditional Bag-of-visual-Words (BoW) concepts. Consider a *vocabulary* $V = [\text{v}_1, \text{v}_2, \ldots, \text{v}_K]$ where each *word-prototype* $\text{v}_k \in \mathbb{R}^{N_d} \; \forall \; k \in \{1, 2, \ldots, K\}$, *i.e.* $K$ word-prototypes where each $\text{v}_k$ is a $N_d$-dimensional vector. Here, the word-prototypes are representative of lower-level visual primitives [49] (*e.g.* SIFT-like features). Under the BoW idea [45], a histogram of word-prototypes (*word-histogram*) can be used as a representation of an image for downstream applications. Through the following insight, we argue that BoW concepts can be leveraged in UniDA, for the problems arising from disjoint label sets.

**Insight 1. (Suitability of BoW concepts for UniDA)** *Word-prototypes represent lower-level visual primitives which are largely unaffected by category-shift in the high-level features. Thus, subsidiary closed-set word-prototype-space alignment assists UniDA with minimal negative-transfer-risk (NTR).*

**Remarks.** We conceptually illustrate this in Fig. 1. Usually, the private categories are somewhat related to the shared categories. Thus, we hypothesize that a set of lower-level visual primitives (or word-prototypes) are capable of representing all categories (even unknown) in the word-prototype-space shared across domains and classes. In Fig. 1, before adaptation, different target features are scattered around the word-prototypes. Then, performing closed-set alignment between the features and word-prototypes can better capture the intrinsic class-structure to support UniDA.

### 3.4 Subsidiary Prototype-space Alignment (SPA)

An obvious question remains: How to realize the closed-set word-prototype-space alignment described in Insight 1? This is crucial because word-prototypes are abstract concepts which need to be explicitly realized. To this end, we propose minor architectural modifications in the backbone.

We insert a block $\psi$ after the backbone $h$ (see Fig. 3). First, a $1 \times 1$-conv. layer converts the $N_d$-dimensional spatial features to $K$-dimensional spatial features (same spatial size as a $1 \times 1$ filter is used with stride 1). We interpret its weight matrix ($V \in \mathbb{R}^{N_d \times K}$) as a vocabulary containing $K$ number of $N_d$-dimensional word-prototypes. The softmax activation performs soft-quantization of the input features $h(x)$ w.r.t. the word-prototypes in $V$.

Consider $h^u(x) \in \mathbb{R}^{N_d}$, the feature vector at a spatial location $u \in \{1, 2, \ldots, U\}$ (where $U$ is the number of spatial locations in the features) and the vocabulary $V = [\text{v}_1, \text{v}_2, \ldots, \text{v}_K]$ where $\text{v}_k$ represents the $k^{\text{th}}$ word-prototype. Then, the soft-quantization (soft equivalent of number of occurrences in a word-histogram) at a spatial location $u$ for the $k^{\text{th}}$ word-prototype $\text{v}_k$ is,

$$[\phi^u(x)]_k = \frac{\exp(\text{v}_k^T h^u(x))}{\sum_{k'} \exp(\text{v}_{k'}^T h^u(x))} \quad (4)$$

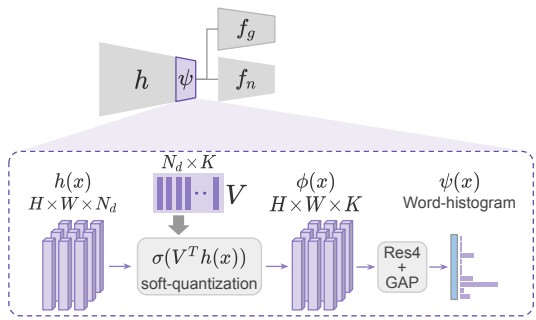

Figure 3: BoW-inspired architecture: $1 \times 1$-conv. layer $V$ (vocabulary) and softmax $\sigma$ soft-quantizes the features in terms of word-prototypes in $V$.

Following this, we employ the remaining Res4-like conv-layers block (as Res3 was chosen for adaptation in Fig. 2) and global average pooling (GAP) to obtain a feature vector $\psi(x) \in \mathbb{R}^K$ from the spatially dense features $\phi(x)$. Thus, we repurpose a simple $1 \times 1$ conv. layer to implement soft word-quantization with the layer weights as the word-prototypes. Now, we introduce a *prototype-alignment-score* (PAS) to further motivate the effectiveness of BoW concepts.

**Prototype-Alignment-Score (PAS).** Consider the backbone feature vector $h^u(x)$ at spatial location $u$. We compute PAS, $\gamma_{PAS}$, as $k$-means loss *i.e.* distance of $h^u(x)$ to the closest word-prototype in $V$,

$$\gamma_{PAS}(h^u(x), V) = 1 - \min_{\text{v}_k \in V} \ell_{cos}(h^u(x), \text{v}_k) \quad (5)$$

Here, $\ell_{cos}$ denotes cosine-distance. Intuitively, $\gamma_{PAS}$ indicates how close the feature vector at $u$ is to one of the word-prototypes in $V$. Since $\phi^u(x)$ is a word-histogram (softmax probabilities), a higher $\gamma_{PAS}$ indicates that the closest prototype would have a much higher contribution in the word-histogram than the others *i.e.* a sparser word-histogram. Based on PAS, we arrive at the following insight.

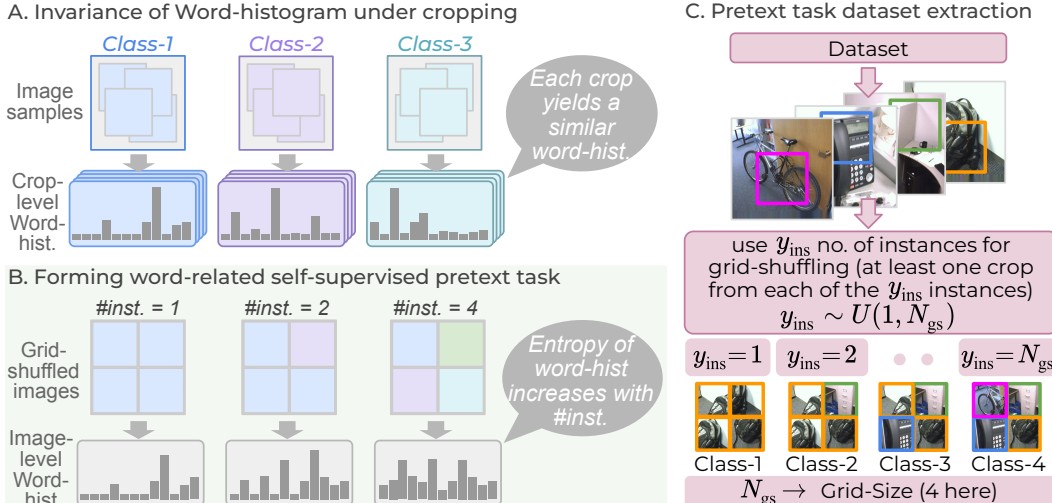

Figure 4: **A.** Different crops of an image yield similar word-histograms. **B.** The word-histogram self-entropy of a grid-shuffled image increases with the number of instances contributing the crops. **C.** Pretext samples are created by grid-shuffling of crops from $y_{ins}$ (pretext label) no. of images.

**Insight 2. (Encouraging Prototype-Alignment for UniDA)** *Since better prototype-alignment implies sparser word-histograms, the word-histogram-space $\phi(x)$ would better represent the intrinsic class structure (including private classes). Thus, by encouraging higher PAS, DA at word-histogram level would exhibit lower negative-transfer-risk as word-prototypes are common across domains.*

**Remarks.** Consider Fig. 1 as an example. The shallower layers would extract generic shapes like rectangles, circles, lines, *etc.* while deeper layers would extract semantic shapes like windows, arms, chassis, etc. Intuitively, the word-histogram space at generic-shape-level cannot be sparse for the object recognition task while sparsity is desirable at deeper layers. Hence, we seek objectives that ensure word-histogram sparsity at a sufficiently high semantic level, catering to the UniDA problem. With this, the pre-classifier features better capture the class-level intrinsic structure that improves UniDA performance. Note that a good intrinsic structure refers to a scenario where individual classes, including private classes, are well clustered in the feature space.

Insight 2 encourages sparser word-histograms and a naive way to enforce this would be a self-entropy minimization objective on the word-histogram vectors $\phi^u(x) \, \forall \, u$. Intuitively, this objective would increase the contributions of the closest word-prototype in $\phi^u(x)$ *i.e.* increase the PAS. Formally,

$$\min_{h,\psi} \mathbb{E}_{x \in \mathcal{D}_s \cup \mathcal{D}_t} \mathbb{E}_u [\mathcal{L}_{em}^u]; \text{ where } \mathcal{L}_{em}^u = -\phi^u(x) \log(\phi^u(x)) \quad (6)$$

However, the self-entropy objective is susceptible to word-level misalignment due to a lack of constraints. For example, different classes may be mapped to the same word-prototype, or different same-class samples may be mapped to distinct word-prototypes. As the target domain is unsupervised with unseen private classes and the word-prototype concepts are abstract, it is very difficult to develop explicit constraints. Thus, we look for implicit constraints via self-supervision to enforce Insight 2.

### 3.5 BoW-based pretext task

We seek a word-related prior or property that can be used to encourage prototype-alignment and better word-prototypes $V$ through a self-supervised pretext task. We hypothesize that different crops from the same image would yield similar word-histograms (Fig. 4A) assuming that all crops are extracted with some part of the object inside the crop. This is reasonable as long as partially out-of-frame crops are not used. Based on this, we provide the following insight to formulate our pretext task.

**Insight 3. (Word-histogram self-entropy prior in grid-shuffled crops)** *Consider a grid-shuffled image where crops from $y_{ins}$ distinct instances are assembled (Fig. 4B). The word-histogram self-entropy of the grid-shuffled image, say $H_e(y_{ins})$, would increase with the number of distinct instances $y_{ins}$ involved in the construction of the grid-shuffled image, assuming a constant grid size $N_{gs}$.*

$$H_e(y_{ins}+1) \geq H_e(y_{ins}) \, \forall \, y_{ins} \in \{1, 2, \ldots, N_{gs}-1\} \quad (7)$$

**Remarks.** Intuitively, combining crops from more number of distinct instances would increase the contributions of distinct word-prototypes (Fig. 4B), thereby increasing word-histogram entropy. This behavior would be consistent when distinct instances also come from distinct task categories, as it ensures minimal overlap of word-prototypes among instances. The assumption of distinct categories for $y_{\text{ins}}$ instances is reasonable when goal task has a large number of categories, which usually holds in common DA benchmarks for object recognition. On the other hand, crops from the same image would only affect a small set of the same word-prototypes and word-histogram entropy can only be marginally affected. While the actual word-histograms may vary with the instances used, the entropy $H_e(y_{\text{ins}})$ for a given number of instances $y_{\text{ins}}$ would lie in a small range.

The pretext task also encourages better prototype-alignment *i.e.* higher PAS. This is because separability of different pretext classes would improve with sparser word-histograms. Concretely, crops from multiple instances (in grid-shuffling) would each have a few distinct word-prototypes with significant contributions and Insight 3 would be better supported. Then, the pretext classes (Fig. 4C) can be easily separated by word-histogram entropy. Note that we use a learnable classifier since these intuitions may not hold at the start of training but pretext objectives can still guide the training.

The pretext task helps avoid word-level misalignment, *i.e.* cases where different classes are aligned with the same word-prototype or different samples of a class are aligned to distinct word-prototypes. Consider Fig. 4A with the worst-case of misalignment where all classes (class-1, class-2, class-3) are represented by the same word-histogram. Then, in Fig. 4B, the image-level word-histograms would be identical for any no. of instances used for patch-shuffling and the pretext task of identifying no. of instances would fail. The above example shows, similar to a proof by contradiction, that the pretext task cannot allow word-level misalignment as misalignment would hurt the pretext task performance.

Based on Insight 3, we construct an entropy-bin for each number of instances $y_{\text{ins}} \in \{1, 2, \ldots, N_{\text{gs}}\}$ for a novel pretext task of entropy-bin classification. Concretely, a pretext classifier $f_n : \mathbb{R}^K \to \mathcal{C}_n$ operates on the output of $\psi$ (Fig. 3) and is trained to predict $y_{\text{ins}}$-class of input grid-shuffled images.

**Procurement of pretext-task samples.** We illustrate this process in Fig. 4C. The same process is followed separately for both source dataset $\mathcal{D}_s$ and target dataset $\mathcal{D}_t$ where goal-task category labels are not required. First, the number of distinct instances $y_{\text{ins}}$ is sampled from a uniform distribution $U(1, N_{\text{gs}})$ and serves as the pretext-label. Next, a batch of $y_{\text{ins}}$ instances is sampled from (say) the source dataset $\mathcal{D}_s$. Now, crops are sampled randomly from the $y_{\text{ins}}$ instances such that each instance contributes at least one crop. Finally, these crops are randomly assembled into the grid-shuffled image denoted by $x_{s,n}$. The same process can be performed for the target dataset to obtain $x_{t,n}$. In summary, a source-pretext dataset $\mathcal{D}_{s,n} = \{(x_{s,n}, y_{\text{ins}}) : x_{s,n} \in \mathcal{X}, y_{\text{ins}} \in \mathcal{C}_n\}$ and a target-pretext dataset $\mathcal{D}_{t,n} = \{(x_{t,n}, y_{\text{ins}}) : x_{t,n} \in \mathcal{X}, y_{\text{ins}} \in \mathcal{C}_n\}$ are extracted.

**Training algorithm.** We aim to demonstrate that our proposed approach is complementary to existing UniDA methods. Thus, we simply add our architecture modification (*i.e.* $\psi$ in Fig. 3) and the pretext classifier head $f_n$ while keeping the other components from an existing method like [36, 23]. Consider an existing UniDA training algorithm denoted by UniDA-Algo$(\mathcal{D}_s, \mathcal{D}_t)$ that trains the backbone $h$ and goal-classifier $f_g : \mathbb{R}^K \to \mathcal{C}_s$. Additionally, we introduce the pretext-task objectives $\mathcal{L}_{s,n} = \mathcal{L}_{ce}(f_n \circ h(x_{s,n}), y_{\text{ins}})$ and $\mathcal{L}_{t,n}$ (defined similarly) for the source and target data respectively (Fig. 5). Here, $\mathcal{L}_{ce}$ denotes the standard cross-entropy loss. Formally, the overall objective is,

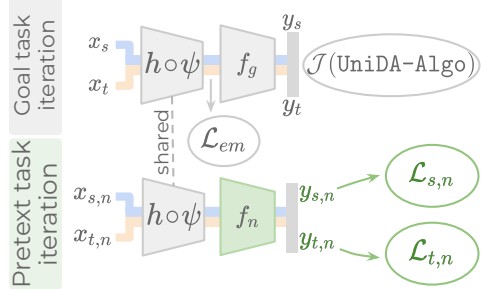

Figure 5: Data samples and objectives for UniDA and pretext task training from Eq. 8.

$$\min_{h, f_g} \mathcal{J}(\text{UniDA-Algo}(\mathcal{D}_s, \mathcal{D}_t)) + \min_{h, \psi, f_n} \left\{ \mathbb{E}_{\mathcal{D}_{s,n}} \mathcal{L}_{s,n} + \mathbb{E}_{\mathcal{D}_{t,n}} \mathcal{L}_{t,n} \right\} + \min_{h, \psi} \mathbb{E}_{\mathcal{D}_s \cup \mathcal{D}_t} \mathbb{E}_u \mathcal{L}_{em}^u \quad (8)$$

Here, $\mathcal{J}(.)$ represents the objective or loss function and the third term is borrowed from Eq. 6. We update only the backbone $h$ and goal-classifier $f_g$ using UniDA-Algo while the backbone $h$ along with the word-prototype layer $\psi$ and pretext-classifier $f_n$ are updated via the pretext-task objectives. Intuitively, the word-prototypes are learnt only through the pretext-task as they are the implicit constraints, as discussed under Insight 2.

Table 1: **Universal DA (UniDA)** on Office-31 and DomainNet benchmarks with HOS metric.

| Method | Office-31 | | | | | | | DomainNet | | | | | | |
|---|---|---|---|---|---|---|---|---|---|---|---|---|---|---|
| | A→D | A→W | D→W | W→D | D→A | W→A | Avg | P→R | R→P | P→S | S→P | R→S | S→R | Avg |
| UAN [48] | 58.6 | 59.7 | 70.6 | 60.1 | 71.4 | 60.3 | 63.5 | 41.9 | 43.6 | 39.1 | 39.0 | 38.7 | 43.7 | 41.0 |
| CMU [10] | 67.3 | 68.1 | 79.3 | 71.4 | 80.4 | 72.2 | 73.1 | 50.8 | 52.2 | 45.1 | 44.8 | 45.6 | 51.0 | 48.3 |
| ROS [3] | 71.3 | 71.4 | 94.6 | 81.0 | 95.3 | 79.2 | 82.1 | 20.5 | 36.9 | 30.0 | 19.9 | 28.7 | 23.2 | 26.5 |
| DANCE [38] | 71.5 | 78.6 | 91.4 | 79.9 | 87.9 | 72.2 | 80.3 | 38.8 | 48.1 | 43.8 | 39.4 | 43.8 | 20.9 | 39.1 |
| I-UAN [46] | 71.5 | 79.4 | 81.5 | 80.7 | 81.0 | 83.0 | 79.5 | - | - | - | - | - | - | - |
| USFDA [16] | 79.8 | 85.5 | 90.6 | 83.2 | 88.7 | 81.2 | 84.8 | - | - | - | - | - | - | - |
| Zhu et al. [53] | **86.1** | 83.2 | 89.8 | **88.0** | 86.7 | **86.6** | 86.7 | - | - | - | - | - | - | - |
| OVANet [36] | 79.4 | 85.8 | **95.4** | 80.1 | 94.3 | 84.0 | 86.5 | 56.0 | 51.7 | 47.1 | 47.4 | 44.9 | 57.2 | 50.7 |
| OVANet+SPA | 80.9 | 85.4 | 92.3 | 82.5 | **97.5** | 82.5 | 86.9 | **61.1** | 51.7 | **47.6** | **48.7** | 45.1 | **58.9** | **52.2** |
| DCC [23] | 78.5 | 88.5 | 79.3 | 70.2 | 88.6 | 75.9 | 80.2 | 56.9 | 50.3 | 43.7 | 44.9 | 43.3 | 56.2 | 49.2 |
| DCC+SPA | 83.8 | **90.4** | 90.5 | 83.1 | 88.6 | 86.5 | **87.2** | 59.1 | **52.7** | **47.6** | 45.4 | **46.9** | 56.7 | 51.4 |

Table 2: **Universal DA (UniDA)** on Office-Home benchmark with HOS metric.

| Method | Office-Home | | | | | | | | | | | | |
|---|---|---|---|---|---|---|---|---|---|---|---|---|---|
| | Ar→Cl | Ar→Pr | Ar→Rw | Cl→Ar | Cl→Pr | Cl→Rw | Pr→Ar | Pr→C | Pr→Rw | Rw→Ar | Rw→Cl | Rw→Pr | Avg |
| UAN [48] | 51.6 | 51.7 | 54.3 | 61.7 | 57.6 | 61.9 | 50.4 | 47.6 | 61.5 | 62.9 | 52.6 | 65.2 | 56.6 |
| CMU [10] | 56.0 | 56.9 | 59.2 | 67.0 | 64.3 | 67.8 | 54.7 | 51.1 | 66.4 | 68.2 | 57.9 | 69.7 | 61.6 |
| I-UAN [46] | 54.1 | 63.1 | 65.2 | 70.5 | 68.3 | 73.2 | 61.9 | 51.8 | 63.8 | 69.8 | 55.6 | 70.7 | 64.0 |
| ROS [3] | 54.0 | 77.7 | 85.3 | 62.1 | 71.0 | 76.4 | 68.8 | 52.4 | **83.2** | 71.6 | 57.8 | 79.2 | 70.0 |
| OVANet [36] | **62.8** | 75.6 | 78.6 | 70.7 | 68.8 | 75.0 | **71.3** | 58.6 | 80.5 | 76.1 | 64.1 | 78.9 | 71.8 |
| OVANet+SPA | 62.0 | 77.7 | **86.3** | 70.0 | 70.1 | 79.3 | 70.0 | 58.8 | 82.5 | 76.8 | 64.0 | 80.5 | 73.2 |
| DCC [23] | 58.0 | 54.1 | 58.0 | 74.6 | 70.6 | 77.5 | 64.3 | 73.6 | 74.9 | **81.0** | 75.1 | 80.4 | 70.2 |
| DCC+SPA | 59.3 | **79.5** | 81.5 | **74.7** | **71.7** | **82.0** | 68.0 | **74.7** | 75.8 | 74.5 | **75.8** | **81.3** | **74.9** |

**Inference.** We discard the pretext-classifier $f_n$ and use only the goal-classifier $f_g$ at inference time. For a fair comparison, we use the same known-unknown demarcation algorithm as `UniDA-Algo`.

## 4 Experiments

**Dataset.** We report results on three different benchmarks. Office-31 [35] contains three domains: DSLR (D), Amazon (A), and Webcam (W). Office-Home [42] is a more difficult benchmark, with 65 classes and 4 domains, Artistic (Ar), Clipart (Cl), Product (Pr), and Real-World (Rw). DomainNet [32] is the largest DA benchmark and the most challenging due to highly diverse domains and huge class-imbalance. Following [10], we use three subsets, namely Painting (P), Real (R), and Sketch (S).

**Evaluation.** We report the mean results of three runs for each experiment. Following [36, 23], target-private classes are grouped into a single unknown class for both Open-Set and UniDA. We report H-score metric (HOS), *i.e.* the harmonic mean of accuracy of shared and target-private samples.

**Implementation Details.** We use two recent prior arts, OVANet [36] and DCC [23], separately as `UniDA-Algo` (Fig. 5). We initialize ResNet50 with ImageNet-pretrained weights and retain other hyperparameters from the original baseline (`UniDA-Algo`). Unless otherwise mentioned, `UniDA-Algo` will be OVANet (for most analysis experiments). We also keep the optimizers, learning rates, and schedulers for both goal task iterations and pretext task iterations same as the baseline. We follow DCC [23] for the dataset and shared-private class splits. See Suppl. for complete details.

### 4.1 Comparison with prior arts

**Open-Set DA.** The benchmark comparisons for OSDA are presented in Table 3 and 4 for Office-Home and Office-31 benchmarks respectively. Our method surpasses all current methods, even those tailored for OSDA [3, 37, 24]. With a 2.2% H-score gain over OVANet [36] and a 4.6% gain over DCC [23], we consistently outperform all OSDA baselines for Office-Home. Similarly, our SPA on top of DCC and OVANet improves on Office-31 by 6.4% and 1.3% over the baselines. Overall, we achieve a superior balance between shared class categorization and private sample identification.

**UniDA.** On the Office-31 benchmark (Table 1), the proposed approach outperforms all other methods in terms of H-score. We improve upon the earlier state-of-the-art methods, DCC [23] and OVANet [36] by 7.0% and 0.4% respectively, again demonstrating a superior balance between shared and private sample identification. Office-Home (Table 2) is a more difficult benchmark, with more

Table 3: **Open-Set DA (OSDA)** on Office-Home benchmark with HOS metric.

| Method | Office-Home | | | | | | | | | | | |
| | Ar→Cl | Ar→Pr | Ar→Rw | Cl→Ar | Cl→Pr | Cl→Rw | Pr→Ar | Pr→Cl | Pr→Rw | Rw→Ar | Rw→Cl | Rw→Pr | Avg |
|---|---|---|---|---|---|---|---|---|---|---|---|---|---|
| STA$_{max}$ [24] | 55.8 | 54.0 | 68.3 | 57.4 | 60.4 | 66.8 | 61.9 | 53.2 | 69.5 | 67.1 | 54.5 | 64.5 | 61.1 |
| OSBP [37] | 55.1 | 65.2 | 72.9 | **64.3** | 64.7 | 70.6 | 63.2 | 53.2 | 73.9 | 66.7 | 54.5 | 72.3 | 64.7 |
| GDA [27] | 59.9 | 67.4 | 74.5 | 59.5 | **66.8** | 70.8 | 60.7 | **58.4** | 70.9 | 65.6 | 61.3 | 73.8 | 65.8 |
| ROS [3] | **60.1** | 69.3 | 76.5 | 58.9 | 65.2 | 68.6 | 60.6 | 56.3 | **74.4** | **68.8** | 60.4 | 75.7 | 66.2 |
| OVANet [36] | 58.4 | 66.3 | 69.3 | 60.3 | 65.1 | 67.2 | 58.4 | 52.4 | 68.7 | 67.6 | 58.6 | 66.6 | 63.3 |
| OVANet+SPA | 59.4 | 67.9 | 75.3 | 62.7 | 65.6 | 70.2 | 61.4 | 54.2 | 71.3 | 68.3 | 58.3 | 71.9 | 65.5 |
| DCC [23] | 52.9 | 67.4 | **80.6** | 49.8 | 66.6 | 67.0 | 59.5 | 52.8 | 64.0 | 56.0 | **76.9** | 62.7 | 63.0 |
| DCC+SPA | 55.2 | **76.0** | 79.5 | 56.2 | 66.2 | **74.0** | **64.2** | 52.5 | 72.2 | 63.8 | 74.4 | **77.0** | **67.6** |

Table 4: **Open-Set DA (OSDA)** on Office-31 benchmark with HOS metric.

| Method | Office-31 | | | | | | |
| | A→D | A→W | D→W | W→D | D→A | W→A | Avg |
|---|---|---|---|---|---|---|---|
| ROS [3] | 82.1 | 82.4 | 96.0 | 77.9 | 99.7 | 77.2 | 85.9 |
| CMU [10] | 70.5 | 71.6 | 81.2 | 80.2 | 70.8 | 70.8 | 74.2 |
| DANCE [38] | 74.7 | 82.0 | 82.1 | 68.0 | 82.5 | 52.2 | 73.6 |
| Inheritune [17] | 81.4 | 78.0 | 92.2 | 83.1 | 99.7 | 91.3 | 87.6 |
| OSHT-SC [9] | **92.4** | 91.3 | 95.2 | **90.8** | 96.0 | 89.6 | 92.5 |
| OVANet [36] | 84.9 | 89.5 | 93.7 | 89.7 | 85.8 | 88.5 | 88.7 |
| OVANet+SPA | 89.7 | 90.2 | **96.9** | 82.6 | **99.8** | 86.8 | 91.0 |
| DCC [23] | 87.1 | 85.5 | 91.2 | 85.5 | 87.1 | 84.4 | 86.8 |
| DCC+SPA | 91.7 | **92.3** | 96.0 | 90.0 | 97.4 | **91.5** | **93.2** |

Table 5: **Ablation study** of our components on Office-Home.

| Method | OSDA | UniDA |
|---|---|---|
| OVANet [36] | 63.3 | 71.8 |
| + architecture mod | 63.6 | 71.8 |
| + arch-mod + $\mathcal{L}_{em}$ | 64.1 | 72.2 |
| + our pretext task | 64.9 | 72.8 |
| + all (SPA) | **65.5** | **73.2** |
| DCC [23] | 63.0 | 70.2 |
| + architecture mod | 63.5 | 70.9 |
| + arch-mod + $\mathcal{L}_{em}$ | 64.2 | 71.5 |
| + our pretext task | 66.1 | 73.6 |
| + all (SPA) | **67.6** | **74.9** |

private classes than the shared classes (55 vs. 10). Our approach exhibits a stronger capability for the separation of shared and private classes in this extreme circumstance, benefiting from our proposed pretext task. On the Office-Home benchmark, our method combined with DCC yields a 4.7% improvement in H-score and an additional 1.4% gain with OVANet. On the large-scale DomainNet dataset (Table 1), our strategy improves OVANet [36] by 1.5%. This shows that our approach outperforms prior arts in a variety of settings, *i.e.* varying degrees of openness.

### 4.2 Discussion

**a) Ablation study.** We perform a thorough ablation study on Office-Home for the components of our approach (Table 5). First, since our architecture modification presents a small increase in computation, we demonstrate that simply using the arch. mod. only marginally improves the performance (0.2% over OVANet and 0.6% over DCC). Further, simply adding the word-histogram self-entropy loss $\mathcal{L}_{em}$ (Eq. 6) with the arch. mod. also yields fairly low improvements over the baseline (0.4% over OVANet and 1.2% over DCC). Next, we assess the improvement from the proposed pretext task (but without $\mathcal{L}_{em}$) and observe gains of 1.1% over OVANet and 3.2% over DCC. Finally, including $\mathcal{L}_{em}$ with the pretext task gives further gains of 0.5% over OVANet and 1.4% over DCC. Thus, the arch. mod. and $\mathcal{L}_{em}$ independently give marginal gains. However, they give significant improvements combined with our pretext task, underlining the importance of every component.

**b) Evaluating clustering of target-private classes.** To further support Insight 2, *i.e.* private class samples are better clustered with our approach, we apply linear evaluation protocol for the target-private classes on the word-prototype features $\phi(x)$. Here, we use the labels of target-private classes (only for this analysis) to train a linear classifier on the frozen features from $\phi(x)$ and compute the accuracy for target-private classes (Table 6). We observe a significant gain (+5.6%) over the baseline OVANet, which indicates that the target-private classes are better clustered with SPA.

Table 6: Target-private accuracy with linear evaluation on frozen $\phi(x)$ for UniDA on Office-Home.

| Method | Ar→Cl | Cl→Pr |
|---|---|---|
| OVANet + arch-mod | 70.7 | 80.8 |
| + Ours | **74.9** | **87.8** |

**c) Correlation between UniDA performance, pretext task performance, PAS and NTR.** For UniDA on Office-Home, we study the correlation between goal task performance (HOS), pretext task performance, negative-transfer-risk (NTR) and prototype-alignment-score (PAS) (averaged over $\mathcal{D}_t$) in Fig. 6C. We observe that goal task performance is positively correlated with both pretext task

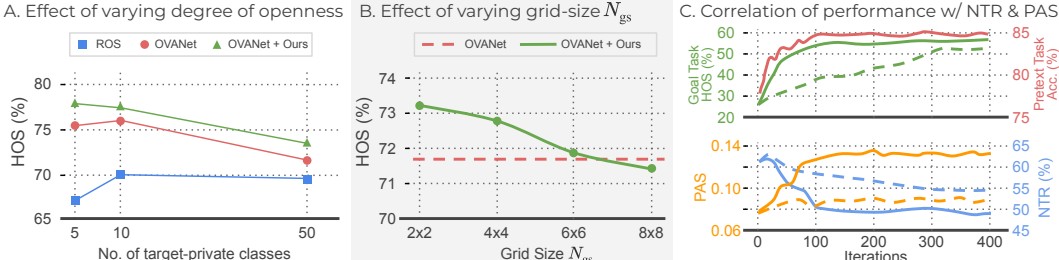

Figure 6: For UniDA on Office-Home, **A.** We evaluate the effect of varying openness *i.e.* no. of target-private classes (Sec. 4.2d), **B.** We report the effect of varying grid-size $N_{gs}$ (Sec. 4.2e), **C.** We study the correlation of goal and pretext task performance with PAS and NTR (Sec. 4.2b). Dashed curves represent experiment performed without our pretext task objectives (as a baseline).

performance and PAS, as in Insight 2. Further, goal task performance is negatively correlated with NTR which supports Insight 1. Further, compared to the baseline OVANet [36] (dashed curves), our OVANet+SPA (solid curves) achieves better and faster convergence (~100 vs. ~300 iterations). Further, we also observe that PAS does not change much when our pretext task is not employed.

**c) Comparison with other pretext tasks.** We compare the effectiveness of our word-related pretext task, for Open-Set DA and Universal DA on Office-Home, with existing pretext tasks in Table 7. Note that we include our EM loss and architecture modification with each of the pretext tasks for a fair comparison. First, we compare with dense-output based pretext tasks like colorization [52, 22] and inpainting [31]. We observe marginal or no improvements with these tasks because the output spaces of these tasks are usually domain-dependent (*e.g.* inpainting of rainy scenes must consider rain, a domain-specific factor), which hinders the adaptation. Next, we consider non-dense classification tasks like rotation [26, 15], jigsaw [5] and patch-location [41]. While these tasks achieve some marginal gains, our proposed pretext task outperforms them as it effectively aids UniDA and Open Set DA by encouraging prototype-alignment (Insight 2).

Table 7: Comparisons with other pretext tasks on Office-Home.

| Method | OSDA | UniDA |
|---|---|---|
| OVANet + arch-mod | 63.6 | 71.8 |
| + colorization | 63.5 | 71.8 |
| + inpainting | 63.7 | 72.0 |
| + jigsaw | 63.8 | 72.0 |
| + patch-loc | 64.0 | 72.1 |
| + rotation | 64.3 | 72.4 |
| + Ours | **65.5** | **73.2** |

**d) Effect of varying degree of openness.** Fig. 6A studies the effect of altering the degree of openness for UniDA in Office-Home. Following [36], we report the average results over five scenarios to cover a variety of domains while varying the amount of unknown classes. Our approach exhibits a low sensitivity to the degree of openness and consistently outperforms other baselines.

**e) Effect of grid-size $N_{gs}$ used in grid-shuffling.** For UniDA on Office-Home, we report a sensitivity analysis for grid-size $N_{gs}$ (Fig. 6B) which controls the number of entropy-bins *i.e.* number of pretext task classes. We observe a marginal decrease in the performance as the grid-size increases, but it outperforms the baseline (OVANet) across a wide range of grid-sizes (2x2 to 6x6). Mitsuzumi et al. [27] show that, as grid-size increases beyond 4x4, shuffling the grid patches significantly alters domain information. Thus, beyond 4x4, performance drops as more domains are introduced which make adaptation difficult. Another reason is that the pretext task itself becomes more difficult as the grid-size is increased. Based on this, we choose grid-size 2x2 ($N_{gs}=4$) for our experiments.

## 5 Conclusion

In this work, we address the problem of Universal DA via closed-set Subsidiary Prototype-space Alignment (SPA). First, we uncover a tradeoff between negative-transfer-risk and domain-invariance at different layers of a deep network. While a balance can be struck at a mid-level layer, we draw motivation from Bag-of-visual-Words (BoW) to introduce explicit word-prototypes followed by word-histogram based classification. We realize the closed-set SPA through a novel word-histogram based pretext task operating in parallel with UniDA objectives. Building on top of existing UniDA works, we achieve state-of-the-art results on three UniDA and Open-Set DA benchmarks.

**Acknowledgements.** This work was supported by MeitY (Ministry of Electronics and Information Technology) project (No. 4(16)2019-ITEA), Govt. of India.

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
