# Supplementary: Subsidiary Prototype Alignment for Universal Domain Adaptation

**Jogendra Nath Kundu**[1*] **Suvaansh Bhambri**[1*] **Akshay Kulkarni**[1*] **Hiran Sarkar**[1]
**Varun Jampani**[2] **R. Venkatesh Babu**[1]

[1]Indian Institute of Science [2]Google Research

## Appendix

In this appendix, we provide more details of our approach, extensive implementation details, additional analyses, limitations and potential negative societal impact. Towards reproducible research, we will publicly release our complete codebase and trained network weights on our webpage.

This supplementary is organized as follows:

- Section A: Notations (Table 1)
- Section B: Limitations
- Section C: Potential societal impact
- Section D: Implementation details
  - Baseline details
  - Compute requirements
  - Miscellaneous details (Fig. 1)
- Section E: Analysis (Table 2, 3)

## A    Notations

We summarize the notations used throughout the paper in Table 1. The notations are listed under 5 groups *i.e.* models, datasets, samples, spaces and measures.

## B    Limitations

The proposed approach may be unsuitable for datasets with very less number of classes. When number of classes are low, our Insight 3 (main paper) may not hold, making the pretext task very difficult to learn. This may negatively impact the goal task performance as the backbone is shared between the two tasks. While this is a limitation of the proposed implementation, a possible solution could be to merge some entropy-bins through clustering techniques to form more discriminative pretext classes. This limitation can also arise in case of large class-imbalance in the data, as this would also lead to overlapping entropy-bins, and a similar bin-merging solution may be used.

## C    Potential societal impact

Our findings may be used to train deep neural networks with minimal supervision by transferring knowledge from supplementary datasets. On many datasets with a large amount of annotated data, such as ImageNet, modern deep networks surpass humans [5]. In many cases where such large-scale

---

*equal contribution

36th Conference on Neural Information Processing Systems (NeurIPS 2022).

Table 1: **Notation Table**

| | Symbol | Description |
|---|---|---|
| Models | $h$ | Backbone feature extractor |
| | $f_g$ | Goal task classifier |
| | $f_n$ | Pretext task classifier |
| | $\psi$ | BoW-inspired block |
| Datasets | $\mathcal{D}_s$ | Labeled source dataset |
| | $\mathcal{D}_t$ | Unlabeled target dataset |
| | $\mathcal{D}_{s,n}$ | Pretext source dataset |
| | $\mathcal{D}_{t,n}$ | Pretext target dataset |
| Samples | $(x_s, y_s)$ | Labeled source sample |
| | $x_t$ | Unlabeled target sample |
| | $(x_{s,n}, y_{\text{ins}})$ | Pretext source sample |
| | $(x_{t,n}, y_{\text{ins}})$ | Pretext target sample |
| Spaces | $\mathcal{X}$ | Input space |
| | $\mathcal{Z}$ | Backbone feature space |
| | $\mathcal{C}_s$ | Source goal task label set |
| | $\mathcal{C}_t$ | Target goal task label set |
| | $\mathcal{C}_n$ | Pretext task label set |
| Measures | $\gamma_{NTR}$ | Negative-Transfer-Risk |
| | $\gamma_{DIS}$ | Domain-Invariance-Score |
| | $\gamma_{PAS}$ | Prototype-Alignment-Score |

related datasets are accessible, our proposed approach can be a proxy to supervision in the target data. Our approach has a favorable impact as it can reduce the data collection effort for data-intensive applications. This might make technology more accessible to organizations and individuals with limited resources. It can also aid applications where data is protected by privacy regulations and hence difficult to collect. The negative consequences might include making these systems more available to organizations or individuals who try to utilize them for illegal purposes. Our system is also vulnerable to adversarial attacks and lacks interpretability, as do all contemporary deep learning systems. While we demonstrate increased performance compared to the state-of-the-art, negative transfer is still possible in extreme cases of domain-shift or category-shift. Thus, our technique should not be employed in critical applications or to make significant decisions without human supervision.

# D  Implementation details

Here, we describe the implementation details excluded from the main paper due to the page limit.

## D.1  Baseline details

**OVANet.** Following prior works [17, 20], we use ResNet50 [6] as our backbone network, which has been pre-trained on ImageNet [13]. We add a new linear classification layer to replace the previous one. We use inverse learning rate decay scheduling to train our models, as described in [17]. We set the weight for entropy minimization loss, $\lambda = 0.1$ for all the settings. The value is calculated by the outcome of Open-Set DA for Office-31 (Amazon to DSLR) following [17]. For all experiments, the source and target batch size is 36. The starting learning rate for new layers is set to 0.01 and for backbone layers to 0.001. Our method is implemented with PyTorch [11].

**DCC.** We use ResNet50 [6] as the backbone, pretrained on ImageNet [13]. The classifier is made up of two linear layers, following [20, 3, 16, 2]. We use Nesterov momentum SGD to optimize the model, which has a momentum of 0.9 and a weight decay of 5e-4. The learning rate decreases by a factor of $(1 + \alpha \frac{i}{N})^{-\beta}$, where $i$ and $N$ represent current and global iteration, respectively, and we set $\alpha = 10$ and $\beta = 0.75$. We use a batch size of 36 and the initial learning rate is set as 1e-4 for Office-31, and 1e-3 for Office-Home and DomainNet. We use PyTorch for implementation.

Table 2: **Open-Set DA (OSDA) on Office-31** with mean and std. deviation over 3 runs. We compare our method with RTN [10], DANN [4], ATI-$\lambda$ [1], OSBP [16], STA [9], InheriT [7], DCC [8].

| Method | A→W | | A→D | | D→W | | W→D | | D→A | | W→A | | Avg | |
|---|---|---|---|---|---|---|---|---|---|---|---|---|---|---|
| | OS | OS* | OS | OS* | OS | OS* | OS | OS* | OS | OS* | OS | OS* | OS | OS* |
| RTN | 85.6±1.2 | 88.1±1.0 | 89.5±1.4 | 90.1±1.6 | 94.8±0.3 | 96.2±0.7 | 97.1±0.2 | 98.7±0.9 | 72.3±0.9 | 72.8±1.5 | 73.5±0.6 | 73.9±1.4 | 85.4 | 86.8 |
| DANN | 85.3±0.7 | 87.7±1.1 | 86.5±0.6 | 87.7±0.6 | 97.5±0.2 | 98.3±0.5 | 99.5±0.1 | 100.0±.0 | 75.7±1.6 | 76.2±0.9 | 74.9±1.2 | 75.6±0.8 | 86.6 | 87.6 |
| ATI-$\lambda$ | 87.4±1.5 | 88.9±1.4 | 84.3±1.2 | 86.6±1.1 | 93.6±1.0 | 95.3±1.0 | 96.5±0.9 | 98.7±0.8 | 78.0±1.8 | 79.6±1.5 | 80.4±1.4 | 81.4±1.2 | 86.7 | 88.4 |
| OSBP | 86.5±2.0 | 87.6±2.1 | 88.6±1.4 | 89.2±1.3 | 97.0±1.0 | 96.5±0.4 | 97.9±0.9 | 98.7±0.6 | 88.9±2.5 | 90.6±2.3 | 85.8±2.5 | 84.9±1.3 | 90.8 | 91.3 |
| STA | 89.5±0.6 | 92.1±0.5 | 93.7±1.5 | 96.1±0.4 | 97.5±0.2 | 96.5±0.5 | 99.5±0.2 | 99.6±0.1 | 89.1±0.5 | 93.5±0.8 | 87.9±0.9 | 87.4±0.6 | 92.9 | 94.1 |
| InheriT | 91.3±0.7 | 93.2±1.2 | 94.2±1.1 | 97.1±0.8 | 96.5±0.5 | 97.4±0.7 | 99.5±0.2 | 99.4±0.3 | 90.1±0.2 | 91.5±0.2 | 88.7±1.3 | 88.1±0.9 | 93.4 | 94.5 |
| DCC | 93.8±1.0 | 99.4±1.1 | 90.7±1.1 | 95.6±0.9 | 96.9±0.5 | 98.4±0.7 | 95.7±0.2 | 98.4±0.1 | 92.5±0.5 | 96.6±0.4 | 94.5±2.1 | 96.3±1.8 | 94.0 | **97.5** |
| +SPA | 96.1±0.5 | 97.0±1.4 | 96.2±1.0 | 97.0±0.3 | 96.0±0.1 | 96.0±0.4 | 99.5±0.2 | 100.0±.0 | 89.2±1.0 | 89.0±0.1 | 91.9±0.9 | 92.0±0.7 | **94.8** | 95.2 |

**Existing code used.**
- OVANet [15]: https://github.com/VisionLearningGroup/OVANet (MIT license)
- DCC [17]: https://github.com/Solacex/Domain-Consensus-Clustering (MIT license)
- PyTorch [11]: https://pytorch.org/ (BSD-style license)

**Existing datasets used.**
- DomainNet [12]: http://ai.bu.edu/M3SDA (Fair use notice)
- Office-Home [19]: https://www.hemanthdv.org/officeHomeDataset.html (Fair use notice)
- Office-31 [14]: https://www.cc.gatech.edu/~judy/domainadapt (open source)

## D.2 Compute requirements

For our experiments, we used a local desktop machine with an Intel Core i7-6700K CPU, a single Nvidia GTX 1080Ti GPU and 32GB of RAM.

## D.3 Miscellaneous details

**Negative-Transfer-Risk (NTR).** We introduce a *negative-transfer-risk* (NTR) $\gamma_{NTR}(h)$ for a given feature extractor $h : \mathcal{X} \to \mathcal{Z}$, where $\mathcal{Z}$ is an intermediate feature-space. First, the standard linear evaluation protocol [18] from transfer learning and self-supervised literature is applied on the feature extractor where a linear classifier $f : \mathcal{Z} \to \mathcal{C}_s$ is trained on the feature $h$ with the labeled source data. Next, following [17], NTR is computed as the known-unknown classification accuracy using a fixed entropy threshold $\rho$ on the linear classifier prediction as:

$$\gamma_{NTR}(h) = \underset{(x, y_{\text{unk}}) \sim \mathcal{D}_t}{\mathbb{E}} \mathbb{1}\left(H_t(f \circ h(x), \rho) = y_{\text{unk}}\right) \text{ where } H_t(f \circ h(x), \rho) = \begin{cases} 1; & H(f \circ h(x)) > \rho \\ 0; & \text{otherwise} \end{cases} \tag{1}$$

where $f = \arg\min_{f'} \mathbb{E}_{(x_s, y_s) \in \mathcal{D}_s} \text{CE}(f' \circ h(x_s), y_s)$ is the learned source classifier on features from $h$. Here, $H(.)$ computes self-entropy, $\rho$ is a fixed entropy threshold, $\log(|C_s|)/2$, where $|C_s|$ represents the number of classes, following [17]. CE represents the standard cross-entropy loss, and $y_{\text{unk}}$ represents known-unknown label (0 for known, 1 for unknown). We access the known-unknown labels $y_{\text{unk}}$ for a subset of target data only for analysis (not for training).

**Pretext dataset procurement.** We illustrate more examples in Fig. 1, based on the procedure given under Insight 3 and in Fig. 4C (main paper).

# E Analysis

**Variance across different seeds.** We highlight the significance of our results by reporting the mean and standard deviation of OS (overall accuracy) and OS* (known classes accuracy) over 3 runs with different random seeds in Table 2 for Open-Set DA on Office-31. We observe low variance with significant performance gains over the baseline.

Table 3: Computational complexity analysis for the BoW-inspired architecture modification.

| | MACS (G) | Params (M) | UniDA |
|---|---|---|---|
| OVANet [15] | 4.120 | 23.661 | 71.8 |
| + arch-mod | 4.223 | 25.686 | 72.0 |

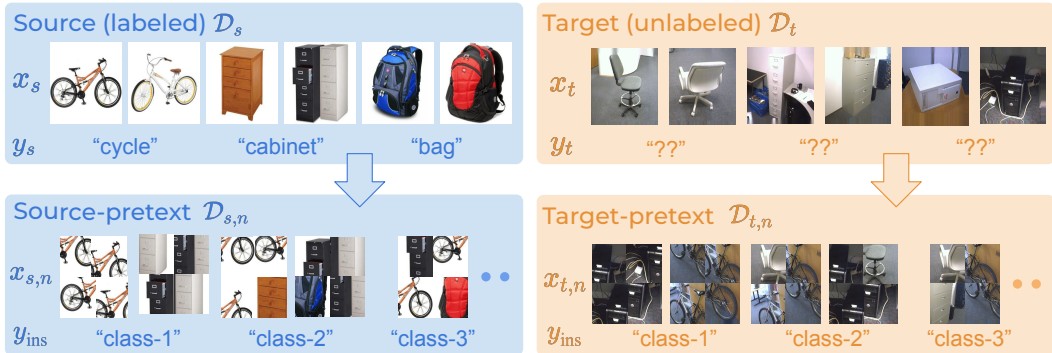

Figure 1: Given labeled source $\mathcal{D}_s$ and unlabeled target $\mathcal{D}_t$ datasets, we construct the source-pretext dataset $\mathcal{D}_{s,n}$ and the target-pretext dataset $\mathcal{D}_{t,n}$ by grid-shuffling of image crops from multiple instances. The pretext task label $y_{\text{ins}}$ is the number of distinct instances contributing image crops.

**Computational complexity comparison.** Table 3 provides the details of the computational overhead caused by the extra parameters added in the BoW-inspired architecture modification (Sec. 3.4 of the main paper). While ~2M additional parameters are required, there is only a marginal increase in the MACS (number of multiply-accumulate operations). Further, simply adding the architecture-modification only marginally improves UniDA (also shown in Table 5, Sec. 4.2a of the main paper).