# OpenReview forum: "Subsidiary Prototype Alignment for Universal Domain Adaptation"
_NeurIPS.cc/2022/Conference — NeurIPS 2022 Accept_

### Official Review · Reviewer_6Jjx · 2022-07-06

**Rating:** 6
**Confidence:** 4
**Soundness:** 3 good
**Presentation:** 2 fair
**Contribution:** 3 good

**Summary:**

In this paper, the authors tackle the problem of universal domain adaptation where the source domain consists of fully-labeled images and unknown category images while the target domain consists of unlabeled images of known and potentially unknown categories. To tackle this problem, they first study the trade-off relation between the negative transfer risk (NTR), i.e., class-level misalignment between the shared and private classes, and domain invariance. Motivated by the trade-off between negative transfer risk and domain invariance, they propose to align mid-level features of deep neural networks. To this end, they propose to align bag-of-word style features with self-entropy regularization to encourage the sparser BoW features. They further regularize the model training by adding a pretext task of entropy-bin classification on the grid-shuffled images from multiple instances. Plugged into the existing UniDA methods, the proposed method shows favorable performance on universal and open-set domain adaptation settings on three public benchmarks.

**Questions:**

Please address all of my concerns in the weaknesses part above. Also, I do have a few more questions for the authors here.

Q1: Why the NTR in Eq (1) does measure negative transfer risk? Basically, it is the accuracy of the pseudo labeling of the shared-private label using entropy thresholding. If we have a high NTR value, we know what are known samples and what are unknown samples. Intuitively it would reduce class-level misalignment.

Q2: In L142-144, the authors claim the domain adaptation should be performed at a deeper layer to encourage domain invariance. However, Fig 2. Shows deeper layers show lower DIS. What is a reasonable conclusion here? Should we more align the deeper layers with lower DIS as they are less domain invariant? Or should we more align the shallower layers with higher DIS as they are domain more promising?

Q3: How do we train a linear domain classifier to measure DIS?

Q4: In Table 6, are the other pretext baselines equipped with the proposed BoW-style architecture modification and L_{em} as well? If not, we need to compare with “our pretext task” baseline in Table 5 (64.9% and 72.8%) instead of full SPA (65.5% and 73.2%).

Q5: In L271-272, why the word prototypes are learnt only through the pretext task? The main task classifier is also attached on top of the word-histogram feature as shown in Fig. 3.



**Limitations:**

The authors did not really address the limitations of their work. Please describe what the failure modes of the proposed method are and when the failures happen. They have not discussed the potential negative societal impact of their work.

**Strengths And Weaknesses:**

The strengths of this paper are as follows.
S1: The problem tackled in this work, universal domain adaptation is challenging and interesting.
S2: Aligning mid-level features for the task of UniDA seems reasonable. Intuitively, high-level features are often specific to classes, therefore prone to category-shift. In contrast, low-level features are robust to category shift. However, they are prone to domain-shift. Therefore, intuitively mid-level features such as BoW could be a good balance between category-shift and domain-shift.
S3: The proposed method shows favorable performance on three public benchmarks. The proposed components, the BoW-style architecture, entropy regularization, and pretext task, contribute to the performance.

However, this paper has several weaknesses as well.
W1: My biggest concern is the study on the relation between NTR and DIS (domain invariance score). Fig. 2 shows Res1 block has the lowest NTR and highest DIS. Compared to Res1 block, Res3 block shows higher NTR and lower DIS. If NTR and DIS are the only criteria, why do we need to use Res3 block feature for alignment? Why we cannot align Res2 and Res1 as well?

W2: The proposed architecture modification does not seem to align the mid-level feature only. By minimizing L_{em} we align the mid-level feature. However, minimizing L_{s,n} and L_{t,n} also align the high-level feature as f_n is attached on top of Res4 and GAP. This is a self-contraction with the claim made from Fig. 2.

W3: The improvement from each component is not very surprising. BoW-style mid-level feature + L_{em} only improves 0.4 ~ 0.5 points on top of OVANet. The results make me doubt that the proposed BoW-style feature alignment is really optimal. In addition, the proposed pretext task improves only 0.4 ~ 0.6 points compared to the rotation task.

W4: Presentation quality should be improved. In general, the paper is not straightforward to read. For example, it is hard to understand why NTR in eq. (1) measures negative transfer risk. Another confusion is that do we need to more align the layers with higher DIS or layers with lower DIS.

---

> ### Author Response · Authors · 2022-08-02
> **Response to Reviewer 6Jjx**
>
> We thank the reviewer for the constructive, detailed and insightful feedback. We appreciate that the reviewer finds our work intuitively reasonable with a challenging and interesting problem setting and favorable results. We address the reviewer's concerns below.
>
> * **[W1, Q2] Why not align res1 and res2? Should we align the layers with low DIS more?**
>     * While the DIS of res1 and res2 is higher than that of res3, we prefer adaptation of layers with lower DIS because we intend to improve the DIS of those layers. Further, aligning res1 and res2 may be trivial as they are already fairly aligned (as indicated by their higher domain-invariance-score DIS) and the deeper layers would remain unaligned.
>     * Based on the above argument, it may seem better to align the deepest layer res4 due to its lowest DIS. However, we need to consider the negative-transfer-risk for the category shift as aligning the highly class-specific (high NTR) deeper layers may lead to misalignment. Thus, we choose a mid-level layer res3 as a compromise between DIS and NTR (considering domain-shift and category-shift respectively).
>
> * **[W2] Pretext task aligns deeper layers?**
>     * We thank the reviewer for pointing this out. While the deeper layers are contained within $\psi$, they are not the same as the baseline deeper layers but rather are processing the word-histogram features. We will update it in the revised draft as "res4-like conv-layers" instead of res4.
>     * To further support our hypothesis, we repeated the Fig. 2 post-adapt analysis where only res3 and word-prototype layer $V$ is updated with the pretext task losses and EM loss (while other losses are as in Eq. 8). We observed an almost similar trend as before, only slightly worse than the post-adapt curves in Fig. 2, indicating that alignment of res3 and word-prototype layer are more significant than that of deeper layers. We will update Fig. 2 and corresponding text in the revised draft.
>
> * **[W3] Is BoW-style alignment optimal?**
>     * As mentioned in L216-220, only using the EM loss may be susceptible to word-level misalignment which leads to lesser gains. Our proposed pretext task helps avoid this misalignment.
>     * The last two rows of Table 5 show that adding the EM loss and BoW-style mid-level features along with the pretext task is better (1.4% average gains for DCC+SPA) than simply using the pretext task.
>
> * **[W3] Proposed pretext task only improves 0.4-0.6 points over rotation**
>     * We respectfully disagree as the gains of our pretext task over rotation are 1% (average) on Office-Home as shown in Table 6 and repeated in below table. Note that `(+x.x)` indicates gains over the previous row.
>     * Further, we show an ablation of rotation pretext and our pretext task on DomainNet below. As DomainNet is a large-scale and challenging benchmark, the gains of our pretext task over rotation signify its efficacy. Note that both pretext task experiments have the mid-level SPA layer as well as the EM loss (the same as in Table 6).
>
> |   |  OSDA (Office-Home)  |  UniDA (Office-Home) |  UniDA (DomainNet) |
> |:------:|:-----:|:-----:|:-----:|
> |  OVANet   |     63.6    |     71.8    |     50.7    |
> |  OVANet+rotation    | 64.3 (+0.7) | 72.4 (+0.6) |       51.1 (+0.4)      |
> |  OVANet+our pretext task  | **65.5 (+1.2)** | **73.2 (+0.8)** | **52.2 (+1.1)** |
>
> * **[Q1] Would high NTR reduce class-level misalignment?**
>     * See our common response to reviewers.
>
> * **[Q3] How to train a linear classifier to measure DIS?**
>     * We use the standard $\mathcal{A}$-distance to measure DIS. We briefly describe the process here and will add the same in the revised supplementary. We train a binary classifier with a linear layer where input is the feature vector (after global average pooling) for which DIS is to be measured, similar to prior works. Source and target training samples, including private class samples, are passed through the frozen network being evaluated to obtain the feature vector. And only the domain-label (0 for source, 1 for target) is required for training the linear classifier with conventional CE loss.
>
> * **[Q4] Do other baselines in Table 6 have SPA-architecture modification and EM loss?**
>     * Yes, we had included those for a fair comparison with our proposed pretext task. Note that performance is slightly lower for all baselines if SPA-modification and EM loss are not used.
>
> * **[Q5] Why are word-prototypes learnt only via pretext task?**
>     * As per Eq. 8 and L266-269, the word-prototypes in $\psi$ are updated only through the pretext task losses and EM loss (the parameters are mentioned under the $\min$ term) while the goal task $\mathtt{UniDA-Algo}$ objective updates only the backbone $h$ and goal classifier $f_g$.
>
> * **Limitations and negative societal impact**
>     * We would like to clarify that, as per the checklist (main paper, L467-468) which the reviewer may have missed, we provide limitations and negative societal impact in Suppl. Sec. 2 and 3.

---

> > ### Comment · Reviewer_6Jjx · 2022-08-08
> > **Major concerns still remain unresolved**
> >
> > [W1,Q2]
> > Thanks for the clarification. The main confusion on Fig. 2 was because it requires quite a mental load to interpret. In my understanding, we need to align layers with lower domain invariance and lower negative transfer risk. This is different from typical trade-off figures. It would be much easier for readers if Fig. 2 is revised. E.g., NTR ($\downarrow$), DIS ($\downarrow$). However, the bigger concern still remains. Have the authors validated that aligning res3 feature does not cluster all the private class samples into one cluster? The evidence of xx percent improvement is weak and indirect.
> >
> > [W2]
> > I do not see any evidence here. I do not understand what “res4-like conv-layers” are. I cannot judge the claim “only slightly worse than the post-adapt curves in Fig. 2, indicating that alignment of res3 and word-prototype layer are more significant than that of deeper layers” as no numbers or figures are provided.
> >
> > [W3]
> > In Table 3 of the main paper, “all (SPA)”, not “OVANet + our pretext”, shows 65.5% OSDA, and 73.2% UniDA performance on the OfficeHome. However, in the rebuttal, the authors show 65.5% and 73.2% are OVAnet + our pretext task. It is really confusing. Which numbers should we trust? If we trust the rebuttal numbers, L_{em} does not contribute at all on top of the proposed pretext task. If we trust the main paper numbers, the proposed pretext task does not improve much compared to the rotation (64.9% vs. 64.3% and 72.8% vs. 72.4%) which confirms my original comment (W3.  the proposed pretext task improves only 0.4 ~ 0.6 points compared to the rotation task).
> >
> > [Q1, Q3, Q5, Limitations and social impact]
> > Thanks for the clarification. My concerns are resolved.

---

> > > ### Author Response · Authors · 2022-08-09
> > > **Reply to Reviewer 6Jjx's post "Major concerns still remain unresolved"**
> > >
> > > Thanks for your constructive feedback, we will update Fig. 2 as per your suggestions to make it more clear. We address the remaining concerns below.
> > >
> > > * **[R-W1, Q2] Have the authors validated that aligning res3 feature does not cluster all the private class samples into one cluster?**
> > >     * To further support our argument that private class samples are better clustered with our approach, we apply linear evaluation protocol for the target-private classes on the word-prototype features $\phi(x)$. Here, we use the labels of target-private classes (only for this analysis) to train a linear classifier on the frozen features from $\phi(x)$ and compute the accuracy for target-private classes.
> > >     * Due to time constraints, we only show two settings of UniDA on Office-Home in the table below. However, we observe a significant gain (+5.62%) in the target-private accuracy with linear evaluation, which indicates that the target-private classes are better clustered with our proposed method.
> > >
> > > | Target-Private Acc. w/ Linear Evaluation | Ar$\to$Cl | Cl$\to$Pr | Avg. Gain |
> > > |---------------------|:---------:|:---------:|:---------:|
> > > | OVANet + arch-mod   | 70.72     |    80.76  |           |
> > > | OVANet + Ours (all) | 74.91     |    87.81  |   **+5.62**   |
> > >
> > > * **[R-W2] Evidence that alignment of res3 and word-prototype layer are more significant than that of deeper layers**
> > >     * We apologize for the confusion caused by our response. Compared to a baseline without our architecture modification and EM and pretext losses, the res4-like conv layers (same architecture as res4 in the baseline) would process the word-histogram features instead of the usual image features.
> > >     * We apologize for not giving concrete numbers. We provide the same corresponding to an updated Fig. 2 in the two tables below. Even when deeper layers are not updated with our proposed pretext and EM losses, the post-adapt NTR and DIS are only marginally worse (relative to the pre-adapt NTR and DIS).
> > >     * Note that we require lower DIS and lower NTR layer for adaptation in order to improve the DIS while further reducing the NTR.
> > >
> > > | Layer | NTR (pre-adapt) | NTR (post-adapt) | NTR (post-adapt w/o updating deeper layers)|
> > > |--|:--:|:--:|:--:|
> > > | Res1 | 44.2 | 34.8 | 35.5 |
> > > | Res2 | 51.3 | 38.2 | 39.1 |
> > > | Res3 | 66.7 | 54.4 | 55.6 |
> > > | Res4 | 70.1 | 59.4 | 60.0 |
> > > |  FC  | 70.3 | 61.9 | 62.8 |
> > >
> > >
> > > | Layer | DIS (pre-adapt) | DIS (post-adapt) | DIS (post-adapt w/o updating deeper layers) |
> > > |--|:--:|:--:|:--:|
> > > | Res1 | 48.0 | 56.7 | 55.9 |
> > > | Res2 | 40.3 | 45.9 | 44.5 |
> > > | Res3 | 23.8 | 30.6 | 29.9 |
> > > | Res4 | 14.5 | 29.1 | 27.7 |
> > > |  FC  | 13.2 | 29.0 | 27.4 |
> > >
> > > * **[R-W3] Confusions with rebuttal ablation table with rotation pretext task**
> > >     * We apologize for the confusion caused by our response. However, we had mentioned just above the table in response-[W3], "Note that both pretext task experiments have the mid-level SPA layer as well as the EM loss (the same as in Table 6)".
> > >     * We repeat the updated rebuttal table below for clarity. Overall, we observe an average 1% improvement over rotation, even on the large-scale DomainNet benchmark.
> > >     * We also give the ablation where architecture modification and EM loss are not used and observe a drop in performance compared to when they are used. Even here, we observe an average 0.9% gain over rotation. DomainNet results are not reported in this case due to time constraints.
> > >
> > > |   |  OSDA (Office-Home)  |  UniDA (Office-Home) |  UniDA (DomainNet) |
> > > |------|:-----:|:-----:|:-----:|
> > > |  OVANet   |     63.6    |     71.8    |     50.7    |
> > > |  OVANet + arch-mod + EM loss + rotation    | 64.3 (+0.7) | 72.4 (+0.6) |       51.1 (+0.4)      |
> > > |  OVANet + arch-mod + EM loss + our pretext task  | **65.5 (+1.2)** | **73.2 (+0.8)** | **52.2 (+1.1)** |
> > >
> > > |   |  OSDA (Office-Home)  |  UniDA (Office-Home) |
> > > |------|:-----:|:-----:|
> > > |  OVANet   |     63.6    |     71.8    |
> > > |  OVANet + rotation (w/o arch-mod & EM loss)    | 63.9 (+0.3) | 72.0 (+0.2) |
> > > |  OVANet + our pretext task (w/o arch-mod & EM loss)  | **64.9 (+1.0)** | **72.8 (+0.8)** |
> > >
> > > * Thanks for your invaluable questions and comments, which have strengthened our submission. Please let us know if any further clarifications are required, as today is the last day for author-reviewer discussions.

---

> > > > ### Comment · Reviewer_6Jjx · 2022-08-09
> > > > **My concerns are resolved**
> > > >
> > > > Thanks for the clarification. Most of my concerns are resolved by the additional evidence. I am increasing my rating.

---

### Official Review · Reviewer_5kcu · 2022-07-08

**Rating:** 4
**Confidence:** 4
**Soundness:** 2 fair
**Presentation:** 2 fair
**Contribution:** 2 fair

**Summary:**

This paper studies the universal domain adaptation problem. Unlike partial DA or open-set DA, UniDA does not require knowing relation between source and target domain label sets. The authors start by analyzing the behaviors of source-trained classifier on target domain data. Specifically, the authors propose a new metric called negative-transfer-risk (NTR). It measures the class-specificity of the features in a specified layer via a shared-vs-unknown binary classifier. The other metric is the inverse of A-distance. Motivated by the analysis, the authors propose the do the adaptation in the mid-level layer of the network. Inspired by the BOW, the authors design a word-prototypes in the selected mid-layer representation, and regularize such a BOW-like representation via a subsidiary prototype-space alignment. To realize SPA, the authors design a novel word-histogram based pretext task with UniDA objectives. The experiments are conducted on three DA benchmarks, and competitive results are reported.

**Questions:**


* As defined in Eq 1, NTR measures the class-specificity of the features in a source-trained specified layer via a shared-vs-unknown binary classifier. So Fig 2 blue solid curve only suggests that in deeper layers (e.g. Res4/FC), it is easier to tell apart the known/unknown given the feature of the source-domain trained network, compared with the feature extracted from the shallow layers. I am not clear why this observation suggests “(line 131: adaptation should be performed at a shallower layer.)”, since the final DA classification task will be performed in the final deep layer.

    Also, I’m not clear if a higher NTR score is better or the other way around. Intuitively, I think higher NTR the better. While when I see the dashed blue curve (post adapt) in Fig 2, it looks that the lower the better. Please clarify.

* Similarly, DIS score only shows the expressive capability of each layer in terms of domain-invariance. Then the conclusion is drawn that the adaptation should be performed in the mid-level layer, which seems contradictory to the existing multi-level adaptation works such as [ref-1].
[ref-1] Xie et al., Multi-Level Domain Adaptive Learning for Cross-Domain Detection, 2019

* Please conduct an ablation study of applying SPA on different layers and all layers.


**Limitations:**

No potential negative societal impact identified.

**Strengths And Weaknesses:**

Strength
* The studied problem Universal DA is a much practical setting, where much weaker assumption is posed compared with standard, partial or open-set DA.

* The technical detail part are generally well written, such as the design of word-prototype.

* The reported results seem competitive. It shows the benefit by applying SPA to both OVAnet and DCC.

Weakness
* The introduction and motivation of NTR are unclear to me and not easy to follow. For detailed questions, please see the below question section.

* Experiments. To support the key statement of mid-level adaptation, the authors miss an ablation study of applying SPA on different layers and all layers.

* From Office-31, [49] seems the best existing method. I’d enough the authors to apply SPA to [49] as [32,20] to show the generalizability of SPA.

* The proposed SPA is based on convolution operator. As vision transformer is becoming more popular and powerful, the applicability of the proposed method may be limited to ConvNet only.

---

> ### Author Response · Authors · 2022-08-02
> **Response to Reviewer 5kcu**
>
> We thank the reviewer for their constructive feedback. We appreciate that the reviewer finds our work generally well-written with a much practical problem setting and competitive results. We address the reviewer's concerns below.
>
> 1. **Confusion with NTR**
>     * We understand that some confusion arose about NTR (please also see our common response) and we clarify that NTR is a risk, implying that lower NTR is desirable. We tried to directly or indirectly highlight it several times in the paper.
>         * L128-130: "In the context of UniDA, the feature space of the deeper layers for source and target would be more difficult to align due to the disjointness of the source and target label sets. We empirically observe the same, i.e. NTR increases as we go deeper in the model."
>         * Eq. 3 and L157: "optimal tradeoff requires NTR to be less than the threshold $\zeta_n$"
>         * L172: "... assists UniDA with minimal negative-transfer-risk (NTR)"
>     * We will highlight the important phrases in the revised draft for clarity.
>
> 2. **Does adaptation at mid-level layer contradicts multi-level adaptation works?**
>     * Multi-level adaptation works (including [ref-1]) operate only on the closed-set DA setting where there is no category shift. In that scenario, adaptation at multiple deeper layers may be useful as DIS (domain-invariance-score) is higher.
>     * However, in presence of category shift, we need to consider not only DIS but also NTR (negative-transfer-risk). As NTR (risk) is higher for deeper layers, we perform adaptation at mid-level layers with lower NTR.
>     * Thanks for mentioning this, as our work is able to explain why multi-level adaptation methods cannot directly work for UniDA (further supported by the multi-layer ablations in the next answer).
>
> 3. **Ablation of SPA at different and all layers**
>     * We performed analysis experiments of DIS and NTR at different layers presented in Fig. 2 but did not add the already performed ablations at different layers. We thank the reviewer for pointing this out and provide these important ablations below which verify our analysis experiments.
>     * We infer that "all layers" implies applying SPA simultaneously after every res-block. As supported by our analysis, we find that all layers and even combination of layers is suboptimal w.r.t. the mid-level res3 which is a *sweet-spot*. Our paper provides concrete evidence on the existence of this *sweet-spot* and how to better utilize it for UniDA.
>
> |     SPA at    | UniDA (Office-Home) |
> |:-------------:|:----------:|
> | none (OVANet) |    71.8    |
> |      res1     |    72.0    |
> |      res2     |    72.4    |
> |      res3     |    **73.2**    |
> |      res4     |    71.9    |
> |   res2+res3   |    72.6    |
> |   res3+res4   |    72.2    |
> |   all layers  |    71.9    |
>
> 4. **SPA with Zhu et al. [49]**
>     * As the code was not available for [49], we re-implemented their method and report the results on Office-31 in the table below. Note that * indicates results from the re-implementation. We observe gains of 1.5% over [49] indicating the generalizability of our proposed SPA.
>
> |                  | UniDA (Office-31) |
> |------------------|:-----------------:|
> |  Zhu et al. [49] |        86.7       |
> | Zhu et al. [49]\* |        86.0       |
> |    [49]\* + SPA   |  **87.5 (+1.5)**    |
>
> 5. **Applicability of SPA to Transformers**
>     * While the SPA layer contains a 1x1 convolution layer, it can be interpreted as a Linear or fully-connected layer as well, which are also used in transformer architectures. Further, the proposed pretext task is independent of the model architecture.
>     * Hence, our proposed SPA can be easily extended for transformers and we report results on Open-Set DA by using a Vision Transformer (ViT) backbone. Note that `(+x.x)` in table indicates the gains over the previous row. Thanks for pointing this out as it further validates the utility of our technique across different architectures.
>
> |                    | OSDA (Office-Home) |
> |--------------------|:------------------:|
> | OVANet (ResNet backbone) |     63.3     |
> | OVANet (ViT backbone) |  64.5 (+1.2)    |
> |   OVANet (ViT) + SPA  |  **65.6 (+1.1)**    |

---

> > ### Comment · Reviewer_5kcu · 2022-08-09
> > **Post-rebuttal**
> >
> > Thanks for the response. This clarifies my question on NTR and Fig 2. It would be good to improve the current presentation of NTR/Fig.2 by incorporating this clarification discussion.
> >
> > I appreciate the efforts of adding extra experiments on additional ablation and backbone. However, in my opinion, the rebuttal is for clarifying the misunderstanding from reviewers, not extensively improving the experiments. To this end, I'm comfortable only raising my rating to borderline reject.

---

### Official Review · Reviewer_S37P · 2022-07-10

**Rating:** 6
**Confidence:** 5
**Soundness:** 3 good
**Presentation:** 3 good
**Contribution:** 3 good

**Summary:**

This paper addresses universal domain adaptation (UniDA) for object recognition: target sample is classified as either one of the "known" classes or "unknown". This work proposes an add-on strategy that helps improve results of existing approaches, reaching SOTA performance in UniDA.

** Analysis **
For UniDA, one wants to increase the domain-invariant score (DIS) between source/target while trying to alleviate the potential negative-transfer risk (NTR) between "known" and "unknown" classes. However, the two measures are often at odds with each other. Quantifying NTR as known-unknown classification accuracy using entropy threshold (like [17]) and DIS as the $\mathcal{A}-$distance (like [1]), the paper provides an analysis on the NTR/DIS trade-off of resnet features at different depths. Empirically, feature of the res3-block balances NTR and DIS. Based on this finding, res3-block feature is used throughout experiments.

** Proposal 1 **
To make domain alignment possible while avoiding the risk of negative-transfer, the paper proposes to align in the Bag-of-visual-words space. The intuition is that BoW contains visual primitives that are universally shared among known and unknown classes. To this end, res3-block features $h\in\mathbb{R}^{H\times W\times N_d}$ are passed through a soft-assignment module including a vocabulary matrix $V\in\mathbb{R}^{N_d \times K}$ ($K$ is the vocabulary size) and a soft-quantization operation (Eqn. 4). For both source and target samples, one minimizes the BoW-wise entropy of the corresponding soft-assignment tensor $\in\mathbb{R}^{H\times W \times K}$. That is to implement the sparsity assumption of the word-histogram space.

** Proposal 2**
A novel pretext task that help encourage prototype-alignment: a pretext dataset is composed of images stitched by a varying number of random image crops from different images. Pretext task is to predict the number of crops, based on the global word-histogram of the stitched image.

** Experiment **
On several benchmarks (Office-31, DomainNet, OfficeHome), the two proposed strategies complement with existing approaches and improve performance, achieving SOTAs in UniDA.

**Questions:**

- What's unclear to me is the sparsity assumption on the word-histogram space. Could the authors elaborate more the intuitions?

- As discussed in L216-220, there are risks of word-level misalignment when minimizing the BoW-entropy. Could the authors provide some real examples when the misalignment happen? How does the proposed pretext task help mitigate the misalignment risks?

**Limitations:**

Limitations and potential negative societal impact are given in the supplementary material. I find those adequate.

**Strengths And Weaknesses:**

- The paper is well-written, easy to follow. Most arguments are sound with adequate empirical supports; there are only a few concerns (see below).

- Originality: the two proposed strategies are novel in the context of UniDA.

- Extensive ablation studies validate the proposed strategies and experimental choices.

- Complementary to existing approaches. The combinations with either OVANet [32] or OCC [20] obtain SOTA results.

---

> ### Author Response · Authors · 2022-08-02
> **Response to Reviewer S37P**
>
> We thank the reviewer for their valuable feedback. We appreciate that the reviewer finds our work novel and sound with adequate empirical support, well-written, and easy to follow. We address the reviewer's concerns below.
>
> 1. **Sparsity assumption of word-histogram space**
>     * Consider Fig. 1 from our main paper. The shallower layers would extract generic shapes like rectangles, circles, lines, etc. while deeper layers would extract semantic shapes like windows, arms, chassis, etc. Intuitively, the word-histogram space at generic-shape-level cannot be sparse for the object recogition task while sparsity is desirable at deeper layers.
>     * Our entropy and pretext objectives ensure word-histogram sparsity at a sufficiently high semantic level, catering to the UniDA problem. With this, the pre-classifier features better capture the class-level intrinsic structure (L209-212) that improves UniDA performance. Note that a good intrinsic structure refers to a scenario where individual classes, including private classes, are well clustered in the feature space.
>
> 2. **Real examples of word-level misalignment; How does pretext task mitigate word-level misalignment?**
>     * Here, misalignment refers to cases where different classes are aligned with the same word-prototype or different samples of a class are aligned to distinct word-prototypes (L216-220).
>     * Consider Fig. 4A with the worst-case of misalignment where all classes (class-1, class-2, class-3) are represented by the same word-histogram. Then, in Fig. 4B, the image-level word-histograms would be identical for any no. of instances used for patch-shuffling and the pretext task of identifying no. of instances would fail.
>     * The above example shows, similar to a proof by contradiction, that the pretext task objectives cannot allow word-level misalignment since misalignment would hurt the pretext task performance. This is how the pretext task helps mitigate word-level misalignment.
>
> * Thanks for your insightful questions. We will incorporate the responses in the revised draft.

---

> > ### Comment · Reviewer_S37P · 2022-08-09
> > **Thanks**
> >
> > I thank the authors for the clear answers. My concerns have been addressed.
> > S37P

---

### Author Response · Authors · 2022-08-02
**Common Response to Reviewers**

* **[6Jjx-Q1, 5kcu-Q1] Confusion with NTR: Would high NTR reduce class-level misalignment?**
    * The reviewers are correct that high NTR implies we know what are known and what are unknown samples i.e. known and unknown samples are well-separated. However, unsupervised adaptation at a higher NTR layer is more susceptible to misalignment between shared and private (unknown) classes because target-private classes get grouped into a single unknown cluster.
    * In contrast, lower NTR feature space can better represent all the different classes without grouping the target-private classes into a single unknown cluster (i.e. better intrinsic structure). Hence, alignment in this space would better respect the separations of private classes than at a higher NTR feature space, which is necessary to avoid misalignment in UniDA.
    * For example, consider "hatchback" (compact car) and "SUV" (large-sized car) as a shared and target-private class, respectively, in an object recognition task. Before adaptation, at a high NTR layer, hatchback and SUV features would be well-separated as SUV is yet unseen to the source-trained model. However, during *unsupervised* adaptation, the similarities between hatchbacks and SUVs may align the single target-private cluster (containing SUV features) with the hatchback cluster. Due to this, other target-private classes also become closer to this hatchback class which increases the misalignment.
    * In contrast, at a lower NTR layer, the target-private classes (including SUV) would not be grouped together. Hence, misalignment of hatchback and SUV clusters would not disturb other clusters unlike the higher NTR scenario. Our proposed sparsity and pretext objectives would help mitigate misalignment even between similar classes.
    * This is supported by our layer-wise ablations (refer to table in response-3 to Reviewer 5kcu) where adaptation at layers with higher NTR is worse than at res3.
* We thank both reviewers for this insightful question and we will update the revised draft for better clarity on NTR.

---

### Author Response · Authors · 2022-08-08
**Gentle Reminder to Reviewers**

We thank all the reviewers for their interesting questions and constructive feedback. We have carefully addressed each point raised in the reviews and hope that our response is satisfactory. We humbly request you to go through our responses and please let us know if any further clarifications are required.

---

### Meta-Review · Area_Chair_FRju · 2022-08-29

**Recommendation:** Accept
**Confidence:** Less certain

**Metareview:**

This submission deals with universal domain adaptation for object recognition. The authors propose to extend existing strategies with an original and effective complementary strategy, thus achieving SOTA performance in this context. Their first proposal aims to align domains while avoiding the risk of negative-transfer, working in the Bag-of-visual-words space. Their second proposal is a new pretext task which seeks to predict the number of crops on images stitched by a varying number of random image crops. This should favor prototype-alignment.

This submission received diverging ratings. Reviewers have raised several concerns, to which the authors have provided detailed answers. The reviewers appreciated the answers and the additional experiences provided. Following the discussions, the final scores of the reviewers have increased and are clearly positive on this submission, on the express condition that all the improvements discussed are integrated in a very careful way.

The AC agrees that the strengths in this case outweigh the weaknesses, but strongly recommend that all the improvements are fully reflected in the final version.

**Award:**

No

---

### Decision · Program_Chairs · 2022-09-14

Accept